# Decision-Driven Calibration for Cost-Sensitive Uncertainty Quantification

Gregory Canal[1], Vladimir Leung[1], John J. Guerrerio[*2], Philip Sage[1], and I-Jeng Wang[1]

[1]The Johns Hopkins University Applied Physics Laboratory
{greg.canal,vladimir.leung,philip.sage,i-jeng.wang}@jhuapl.edu

[2]Dartmouth College
john.j.guerrerio.26@dartmouth.edu

## Abstract

In recent years, the ability of artificial intelligence (AI) systems to quantity their uncertainty has become paramount in building trustworthy AI. In standard uncertainty quantification (UQ), AI uncertainty is calibrated such that the confidence of its predictions matches the statistics of the underlying data distribution. However, this method of calibration does not take into consideration the direct influence of UQ on the subsequent actions taken by downstream decision-makers. Here we demonstrate an alternate, decision-driven method of UQ calibration that explicitly minimizes the incurred costs of downstream decisions. After formulating decision-driven calibration as an optimization problem with respect to a known decision-maker, we show in a simulated search-and-rescue scenario how decision-driven temperature scaling can lead to lower incurred decision costs.

## 1 Introduction

Artificial intelligence (AI) has become an integral component in decision-making pipelines in many settings across society, ranging from medicine to scientific discovery. Rather than providing only a single predicted output in these systems, it has become common for AI to also supplement their predicted outputs with quantified notions of *uncertainty* describing the level of confidence in predictions, i.e., to perform *uncertainty quantification* (UQ). For a decision-maker taking actions in downstream tasks, only considering the AI's predicted output in isolation runs the risk of discounting critical information about possible sources of ambiguity in the AI's output. In contrast, taking the full UQ distribution into consideration can better enable informed decision-making that trades off the potential costs and benefits of various actions [1]. For instance, a doctor aided by an AI diagnosis system can better weigh the risks of various patient treatment options if the AI provides a confidence level for its diagnosis, only moving ahead with risky treatments if the AI is confident enough.

Rather than serving only as a passive aid, in general we expect the specific UQ outputs generated by the AI to have a significant impact on the actions actually taken by downstream decision-makers consuming these outputs. Continuing with the above example, if the doctor only proceeds with a treatment if the AI's confidence exceeds a particular threshold (e.g., 90%), then the difference of only a few confidence percentage points may have a drastic effect on the patient's outcome. This implies that the selection and tuning of the UQ algorithm (e.g., temperature scaling [2], conformal prediction [3]) can potentially result in different behaviors by the downstream decision-maker [4].

---

[*]Completed during an internship at JHU/APL.

Workshop on Bayesian Decision-making and Uncertainty, 38th Conference on Neural Information Processing Systems (NeurIPS 2024).

Based on these observations, we note that common UQ calibration techniques minimizing distributional metrics such as Expected Calibration Error (ECE) do not necessarily improve decision-making performance in the downstream application of interest, as measured by incurred decision costs. Recognizing the critical role of UQ in the overall decision-making pipeline, in this work we introduce *Decision-Driven Calibration* (DDC) as a means of optimizing UQ parameters to explicitly minimize expected downstream decision costs. Although aspects of decision-aware calibration have been explored in prior work, to our knowledge this work is the first to optimize UQ for minimization of incurred costs with respect to a given decision-cost function and a (potentially non-ideal) decision-maker profile. We demonstrate in a simulated search-and-rescue task how DDC can reduce decision costs in comparison to standard temperature scaling-based calibration.

**Related work** UQ comprises a vast body of work, providing a multitude of methods for learning uncertainty estimates from data [5, 6]; here we specifically focus on prior work leveraging knowledge of the downstream decision process to inform UQ selection and calibration. [7] focuses calibration in output probability regions that have the most impact on diagnostic decision making, but considers only the scenario of an unknown decision cost function in narrow settings. Another line of work introduces the notion of Decision Calibration Error (DCE), measuring the discrepancy between the estimated decision costs incurred by an output probability distribution in comparison to the true costs incurred under the true data distribution [8, 9, 10]. While optimizing UQ to minimize DCE does ensure that decision costs computed under the UQ distribution are reliable estimates of the true expected cost, this is distinct from our goal of optimizing UQ to *minimize the incurred decision costs themselves*. [11] does optimize UQ to minimize downstream decision costs while taking into account a given decision-maker model, but is limited to the case of binary decisions. [12] uses a fixed decision-maker model to optimize conformal prediction for minimizing decision-maker classification error; however, this approach is limited to classification actions. [13] explicitly optimizes UQ to minimize incurred decision-making costs, but the decision space is limited to the decision-maker either accepting the AI's recommendation or solving the task themselves.

## 2 Methods

Let $\mathcal{X}$ denote a data domain (e.g., $\mathcal{X} = \mathbb{R}^d$) with individual examples denoted by $\boldsymbol{x}$. Here we consider the scenario of multi-class classification where each $\boldsymbol{x}$ is associated with a label $y \in 1 \ldots C$, but the concepts demonstrated here could be applied to other settings (e.g., regression). Consider an AI model given by the function $f_\theta \colon \mathcal{X} \to \mathbb{R}^C$ mapping from examples to "logit" vectors $\boldsymbol{z} = f_\theta(\boldsymbol{x})$. In standard classification, the AI model's prediction would be taken as $\widehat{y} = \arg\max_{y'} f_\theta^{(y')}(x)$, where $f^{(i)}$ denotes the $i$th entry of $f$. More generally, logit $\boldsymbol{z}$ can be used to generate a UQ output such as a probability distribution (via the softmax function) or a conformal set. Adopting notation from [4], for generality let $g_\phi \colon \mathbb{R}^C \to \mathcal{U}$ denote a generic UQ function parameterized by $\phi$ and mapping to an uncertainty representation space $\mathcal{U}$. For instance, for softmax class probabilities we have $g \colon \mathbb{R}^C \to \Delta^{C-1}$ given by $g(\boldsymbol{z}) = \sigma(\boldsymbol{z})$ where $\sigma$ is the softmax function.

We suppose that a decision-maker observes $g_\phi(f_\theta(\boldsymbol{x}))$ and chooses an action $a$ from some set of $K$ actions $\mathcal{A}$, noting that it need not be the case that $\mathcal{A} = 1 \ldots C$ or even for $K = C$. Without loss of generality we model the decision-maker as a known conditional probability distribution $\delta \colon \mathcal{U} \to \Delta^{K-1}$ and assume that the action $a$ for data point $\boldsymbol{x}$ is sampled from $\delta(g_\phi(f_\theta(\boldsymbol{x})))$. We assume that taking action $a$ on example $\boldsymbol{x}$ with ground-truth label $y$ incurs a cost of $c(y, a)$, where $c(\cdot, \cdot)$ is a known cost function. For data $\boldsymbol{x}, y$ distributed according to $\mathcal{D}_{X,Y}$, we define the expected decision cost as $C(\phi) = \mathbb{E}_{\boldsymbol{x}, y \sim \mathcal{D}_{X,Y}} \mathbb{E}_{a \sim \delta(g_\phi(f_\theta(\boldsymbol{x})))}[c(y, a)]$. For optimal decision-making (in an average sense), the expected incurred cost $C$ should be as low as possible.

**Decision-agnostic calibration** In typical calibration, the parameters of $g_\phi$ are adjusted such that the UQ output matches the underlying statistics of the data distribution. In practice this is typically achieved by optimizing a scoring rule $\ell$ over a calibration set $S_{\mathrm{cal}}$ sampled from $\mathcal{D}_{XY}$:

$$\phi^* = \arg\min_\phi \frac{1}{|S_{\mathrm{cal}}|} \sum_{\boldsymbol{x}, y \in S_{\mathrm{cal}}} \ell(g_\phi(f_\theta(\boldsymbol{x})), y). \tag{1}$$

In this work we focus on standard Temperature Scaling (TS) applied to a label distribution [2] optimized with Negative Log-likelihood (NLL) as the UQ method and associated calibration criteria.

TS adjusts the model's output probabilities by dividing the logits by a scalar "temperature" parameter $T$ which smooths the probability distribution without altering the ranking of predicted classes. In our notation, this corresponds to a UQ function $g_T(\boldsymbol{z}) = \sigma(\boldsymbol{z}/T)$, where the only UQ parameter $\phi$ to be optimized is the temperature $T$. Letting $g^{(i)}$ denote the $i$th entry of $g$, the NLL loss is given by $\ell_{\text{NLL}}(g_T(\boldsymbol{z}), y) = -\log g_T^{(y)}(\boldsymbol{z})$ which can be optimized in (1) over the calibration set.

**Decision-driven calibration**    While performing standard calibration as in (1) will encourage the UQ outputs $g$ to match the underlying statistics of the data distribution, this method of calibration does not necessarily ensure a reduced downstream decision cost $C$. Instead, as a loss function in (1) we utilize a decision-driven calibration loss, computed as the expected decision cost incurred with respect to a fixed decision-maker $\delta$ (denoting the $i$th entry of $\delta$ by $\delta^{(i)}$):

$$\ell_{\text{DDC}}(g(\boldsymbol{z}), y) = \sum_{a \in \mathcal{A}} \delta^{(a)}(g(\boldsymbol{z}))\, c(y, a). \tag{2}$$

We refer to the optimization of (1) with $\ell_{\text{DDC}}$ as *decision-driven calibration* (DDC). Although DDC assumes that both $\delta$ and $c$ are fixed and known, it is possible to learn $\delta$ from observed decision-maker behavior ([11, 12]), and in many settings numeric costs can be assigned to various action outcomes.

**Decision-maker models**    We consider two possible decision-maker models for optimization in (2) that accept as input a label distribution and generate a decision:

*Smooth Bayes' Optimal:* It is well-known that the average-case optimal action is the *Bayes' optimal* action minimizing the expected incurred cost, i.e., $a^* = \arg\min_a \sum_{y'} p^{(y')} c(y', a)$, where $p^{(y)}$ is the *true* conditional probability of an example having label $y$. Since the decision-maker only has access to $g(\boldsymbol{z})$ and not $p$, they can instead take a Bayes' optimal action with respect to $g(\boldsymbol{z})$ as $\hat{a} = \arg\min_a \sum_{y'} g^{(y')}(\boldsymbol{z}) c(y', a)$, or in vector notation as $\delta = \mathbb{1}_{a=\hat{a}}$, i.e., the one-hot vector at $\hat{a}$. To make this decision-maker differentiable, we apply the softmax approximation $\mathbb{1}_{i=\arg\max_i \boldsymbol{z}^{(i)}} \approx \sigma(\boldsymbol{z})$ and express the $\arg\min$ in $\hat{a}$ as a negated $\arg\max$. This leads to $\delta = \sigma(-\boldsymbol{L}^T g(\boldsymbol{z}))$ where $L$ is the $C \times K$ matrix whose $L[i, j]$ entry is given by $c(i, j)$.

*Proportional:* We model the decision-maker as first selecting a label $y'$ with probability $g^{(y')}(\boldsymbol{z})$, and then taking the optimal action for that class. This corresponds to a probability of action $a$ given by $\sum_{y'} \boldsymbol{A}[y', a]\, g^{(y')}(\boldsymbol{z})$ where $\boldsymbol{A}$ is the $C \times K$ matrix given by $\boldsymbol{A}[i, j] = 1$ for $j = \arg\min_a c(i, a)$ and $\boldsymbol{A}[i, j] = 0$ otherwise. We can represent this decision-maker in vector form as $\delta = \boldsymbol{A}^T g(\boldsymbol{z})$.

## 3   Simulated decision-making

We aim to demonstrate in a realistic decision-making setting how DDC can result in reduced test-time decision costs when applied to TS, in comparison to standard calibration. To do so, we simulate an AI-assisted search-and-rescue scenario where a drone is tasked with patrolling a section of beach and assisting swimmers in distress. We assume the drone will encounter objects belonging to three classes: `boat`, `swimmer without a life jacket`, and `swimmer with a life jacket`. In this example we assume that all swimmers without a life jacket are in distress, while all swimmers with a life jacket are not in distress. Upon encountering a new object, the drone can take one of three actions: continue its route without intervention (*Do Nothing*), "mark" the object to return to later (*Mark Object*), or drop a flotation device and call for help (*Rescue*).

For each class-action pair we assign a nominal numerical "cost" characterizing the quality of each decision, presented in Table 1 and justified as follows: the most severe error is ignoring a swimmer without a life jacket (i.e., in distress). "Marking" these swimmers as

Table 1: Search-and-rescue decision cost matrix ($c(y, a)$).

| Ground-truth class ($y$) | Action ($a$) | | |
| --- | --- | --- | --- |
| | *Do Nothing* | *Mark Object* | *Rescue* |
| boat | 0 | 15 | 50 |
| swimmer w/o life jacket | 100 | 75 | 0 |
| swimmer w/ life jacket | 10 | 5 | 30 |

objects of interest is also a significant error, since it delays rescue efforts. "Rescuing" or "marking" a boat or a swimmer with a life jacket incurs a moderate cost, to reflect wasted effort.

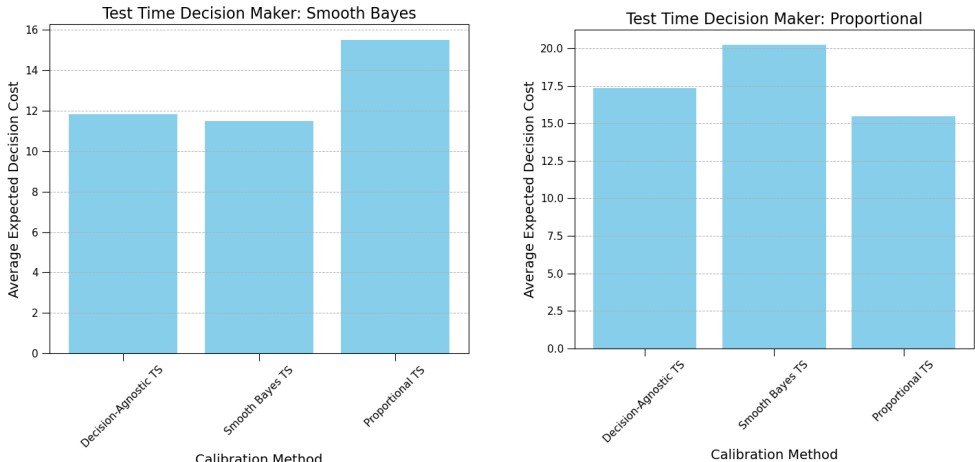

Figure 1: test-time expected decision cost.

**Dataset and model training**  We simulate this decision-making process on the multi-object tracking (MOT) subset of SeaDronesSee [14], a large-scale dataset of overhead aerial drone footage (see example images in Figure 3). This dataset consists of 22 video clips, encompassing a total of ∼50k frames and ∼400k annotations and contains objects in the classes described above.[2] We split SeaDronesSee-MOT as follows: for our "test" set we use the original "valiation" split. To construct a "calibration" set, within each video we partition the original MOT "training" set by allocating the first 75% for model training and the final 25% for calibration. To focus only on the task of classification, we crop each image to the ground-truth bounding boxes of objects in the scene. This results in ∼120k training instances, ∼40k calibration instances, and ∼50k testing instances.

For the AI predictive model $f_\theta$ mapping from cropped images to logits $z$, we train a single linear layer as the classification head on a ViT-B/16 backbone [15]. The ViT backbone was pretrained on the SWAG dataset and fine-tuned end-to-end on ImageNet-1k [16]. After freezing the backbone, we fine-tune the classification head on our custom SeaDronesSee-MOT training split (excluding the calibration set). We used a learning rate of 0.003, Cross-Entropy loss with a class re-weighting scheme from [17], and optimized with AdamW [18]. As a UQ output, we then apply TS to the model Softmax outputs $\sigma(z)$, optimizing over the calibration split using either NLL or DDC calibration loss. Further training details can be found in Appendix A.

**Numerical results**  Our main objective is to compare the average decision cost at test-time incurred by decision-driven calibration TS in comparison to decision-agostic (i.e., standard) TS. For decision-driven calibration, we refer to the calibration method according to the decision-maker assumed during calibration (e.g., Smooth Bayes TS, Proportional TS). For a given temperature $T$ (as optimized by one of the calibration methods above), we measure the incurred cost over a test set $S_{\text{test}}$ as $C(T) = \frac{1}{|S_{\text{test}}|} \sum_{x,y \in S_{\text{test}}} \ell_{\text{DDC}}(g_T(f_\theta(x)), y)$, where $\ell_{\text{DDC}}$ is computed with respect to a test-time decision-maker $\delta_{\text{test}}$ not necessarily equal to the one used during decision-driven calibration (when applicable). In particular, we evaluate all TS methods (Decision-agnostic TS, Smooth Bayes TS, and Proportional TS) against both the Smooth Bayes and Proportional decision-makers (Figure 1). When computing $C(T)$ for temperatures obtained from Smooth-Bayes DDC, we observe a smaller incurred decision cost than Decision-agnostic TS. However, we find that calibrating according to a mismatched Proportional decision-maker incurs a worse cost than decision-agnostic TS. Similarly, when evaluating against a Proportional decision-maker at test-time, Proportional TS incurs a smaller cost than Decision-agnostic TS, which itself incurs a lower cost than Smooth Bayes TS. These results indicate that performing decision-driven calibration with temperature scaling does lead to reduced costs at test time, but a mismatch in decision-maker between calibration and test-time can actually *increase* decision costs.

---

[2]To simplify our decision-making scenario we discard objects of the `empty life jacket` class.

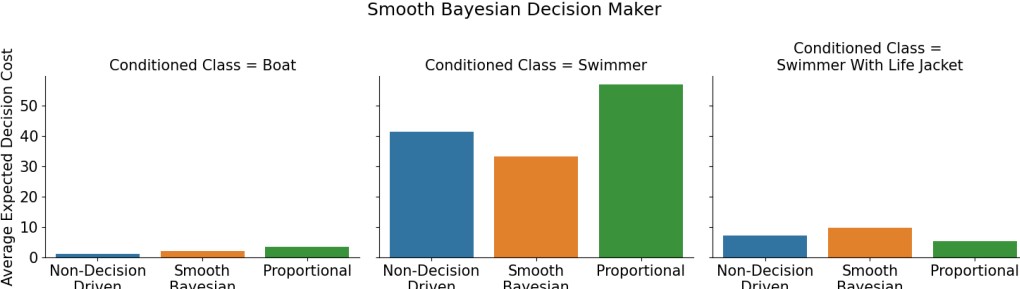

Figure 2: Class-conditional average decision cost at test-time (Smooth Bayes decision maker).

The difference in incurred cost between the various calibration methods becomes even more apparent when decomposing average cost by ground-truth class. In the case of a Smooth Bayes decision-maker, calibrating with respect to the same decision model results in a much lower average cost incurred on the Swimmer class compared to Decision-agnostic and Proportional calibration (Figure 2). This is a desirable outcome in our search-and-rescue scenario, as swimmers without life jackets are the most critical class to protect. Interestingly, a tradeoff is made by slightly *increasing* average cost when conditioned on the other ground-truth classes. This could be attributed to the structure of the cost matrix (Table 1), where taking incorrect actions on a swimmer without a life jacket results in significantly larger penalties than incorrect actions on other classes. Similar observations can be made when examining classwise costs when evaluated with a Proportional decision-maker (Figure 4).

When examining the optimized temperature values for each calibration method, we observe that decision-agnostic temperature scaling and Smooth Bayes temperature scaling actually adjust the temperature in *opposite* directions (i.e., less or greater than 1.0), resulting in opposite effects on model confidence (Figure 5): decision-agnostic temperature scaling *increases* the confidence of model predictions (decreased temperature), while Smooth Bayes temperature scaling *decreases* the confidence of model predictions (increased temperature). Future research will be crucial in understanding how the magnitude of specific action costs and the balance of costs between classes affects decision-driven calibration performance. Additional results comparing the calibration performance of decision-agnostic and decision-driven calibration can be found in Appendix B, including a comparison of test-time ECE and NLL (Figure 6) and reliability diagrams (Figure 7).

## 4  Conclusion

Overall, this work introduces decision-driven calibration as a means of directly optimizing UQ to minimize incurred costs for a known decision-maker, and to our knowledge is the first work to perform this type of calibration on a general decision cost function. We demonstrate the promise of DDC in comparison to standard temperature scaling for reducing incurred costs in a simulated scenario on a large-scale search-and-rescue dataset. Our results also demonstrate the importance of calibrating according to a decision-maker matched to the one taking actions at test-time, and motivates the approach in [11, 12] of accurately estimating a decision-maker before calibration occurs. Important future avenues of work include a theoretical study of the robustness of DDC to decision-maker mismatch or shifts in decision-making behavior, and the utilization of active learning [19] to efficiently estimate a decision-maker model from a minimal number of queries.

## Acknowledgements

Copyright 2024 Carnegie Mellon University and JHU APL. This material is based upon work funded and supported by the Department of Defense under Contract No. FA8702-15-D-0002 with Carnegie Mellon University for the operation of the Software Engineering Institute, a federally funded research and development center. DM24-1545.

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

# A Implementation details

## A.1 Dataset construction

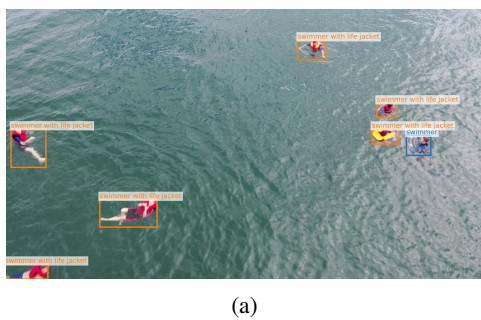 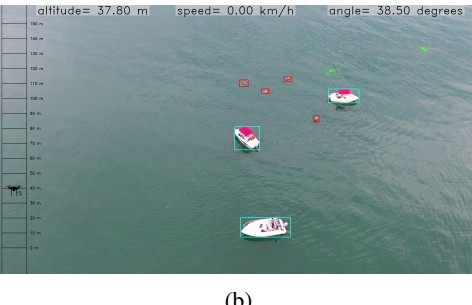

|           (a)            |           (b)           |

Figure 3: Example images from the SeaDronesSee dataset [14]. We generate the data for our experiments by cropping the bounding boxes for the classes `boat`, `swimmer without a life jacket`, `swimmer with a life jacket`.

The original SeaDronesSee-MOT dataset was created by dividing each video clip into three segments: the first 4/7 of each clip serves as the MOT training set, the next 1/7 as the MOT validation set, and the final 2/7 as the MOT test set [14]. We exclude the MOT test set from our analysis because it lacks ground-truth object labels. For our task we focus only on classifying objects in SeaDronesSee rather than needing to detect each object in addition; therefore, we preprocess each scene image by first cropping each object according to its ground-truth bounding box, and treating each cropped bounding box as its own data point $x$ with class label $y$.

We sort each video in the MOT training set by image ID, arranging the frames in sequential order, and allocate the final 25% of frames in each training-set video to the calibration set. We take the original MOT "validation" set as our test set without any modifications. Since the MOT validation set comes after the MOT training set in the sequence of frames, this splitting approach ensures that there is no overlap between the training, calibration, and validation set. Furthermore, by selecting the final 25% of frames in the MOT training set videos for calibration, we also introduce a gap in the sequence between the training set and validation set, which helps create more distinct test examples. We summarize the resulting instance counts for each split in Table 2.

Table 2: Dataset class instances

| Class | Training | Validation | Calibration | Total |
|---|---|---|---|---|
| Boat | 51,334 | 20,715 | 16,763 | 88,812 |
| Swimmer without Life Jacket | 22,877 | 10,107 | 7,305 | 40,289 |
| Swimmer with Life Jacket | 45,330 | 16,856 | 15,845 | 78,031 |
| **Total** | **119,541** | **47,678** | **39,913** | **207,132** |

## A.2 Training details

The classifier model has a total of 2,307 trainable parameters, resulting from the classification layer's reduction of the 768-dimensional embedding vector to 3 dimensions, corresponding to the number of classes. After each training epoch, we evaluate the model on the **calibration set** and chose the best performing model for our experiments. The best performing model came from epoch 56 with 98.259% accuracy. Full training data augmentation details and model hyperparameters are presented in Table 3 and Table 4, respectively.

Table 3: Data augmentation hyperparameters

| Hyperparameter | Value |
|---|---|
| Mixup $\alpha$ | 0.2 |
| Cutmix $\alpha$ | 1.0 |
| Random Augmentation Magnitude | 9 |
| AugMix Severity | 3 |
| Interpolation | Bilinear |
| Random Augmentation Sampling | True |
| Random Augmentation Repetitions | 3 |

Table 4: Model training hyperparameter configuration

| Hyperparameter | Value |
|---|---|
| Backbone | vit_b_16 |
| Pretrained Weights | ViT_B_16_Weights.IMAGENET1K_SWAG_E2E_V1 |
| Input Size | 384x384 |
| Optimizer | AdamW |
| Batch Size | 1024 |
| Training Epochs | 70 |
| $\alpha$ (Learning Rate) | 0.003 |
| $\beta$ (Momentum) | 0.9 |
| $\lambda$ (Weight Decay) | 0.3 |
| Learning Rate Scheduler | Cosine Annealing |
| Learning Rate Warmup Epochs | 20 |
| Learning Rate Warmup Method | Linear |
| Learning Rate Warmup Decay | 0.033 |
| Learning Rate Step Size | 30 |
| $\gamma$ (Learning Rate Decay Factor) | 0.1 |
| Automatic Mixed Precision | True |
| Gradient Clipping Threshold | 1.0 |
| Loss | Cross Entropy |
| Class Reweighting in Loss Function | True |

## A.3 Temperature scaling hyperparameters

As in [20] we use LBFGS to optimize the temperature parameter during both decision-agnostic and decision-driven calibration, using the choice of hyperparameters listed therein. We add a small value $\epsilon$ (1e-6) to the temperature parameter during optimization to prevent numerical instabilities. We summarize these calibration hyperparameters in Table 5.

Table 5: Self-Assessment Hyperparameters - Temperature Scaling

| Hyperparameter | Value |
|---|---|
| Optimizer | LBFGS |
| Maximum Number of Optimizer Steps | 3000 |
| $\alpha$ (Learning Rate) | 0.1 |
| $\epsilon$ | 1e-6 |
| Optimizer Line Search Function | Strong Wolfe |
| Loss | Cross Entropy Loss |
| Initial Temperature Value | 1 |

# B Additional figures

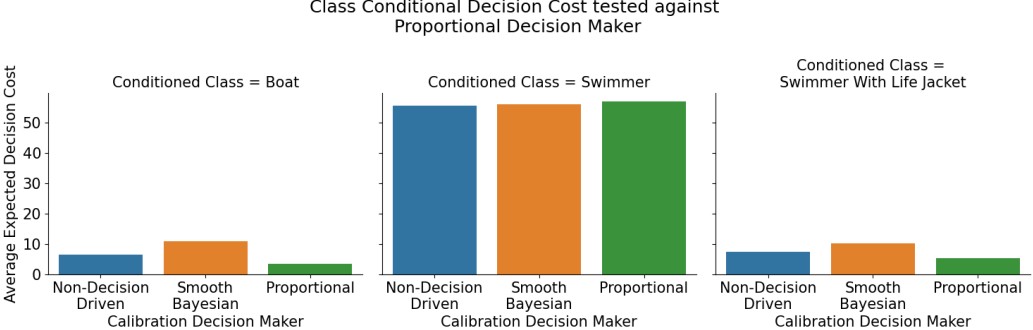

Figure 4: Class-conditional average decision cost at test-time (Proportional decision maker).

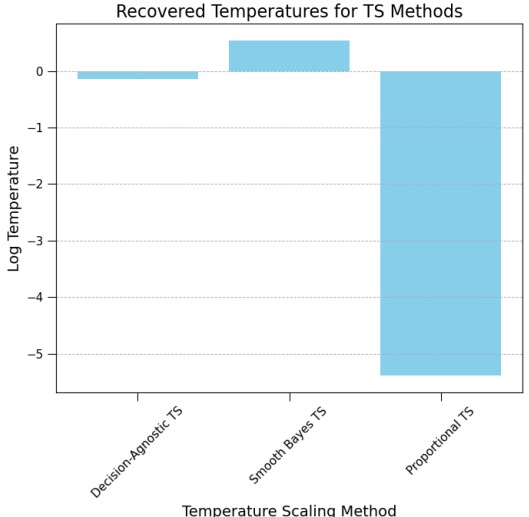

Figure 5: Temperatures selected by each TS method.

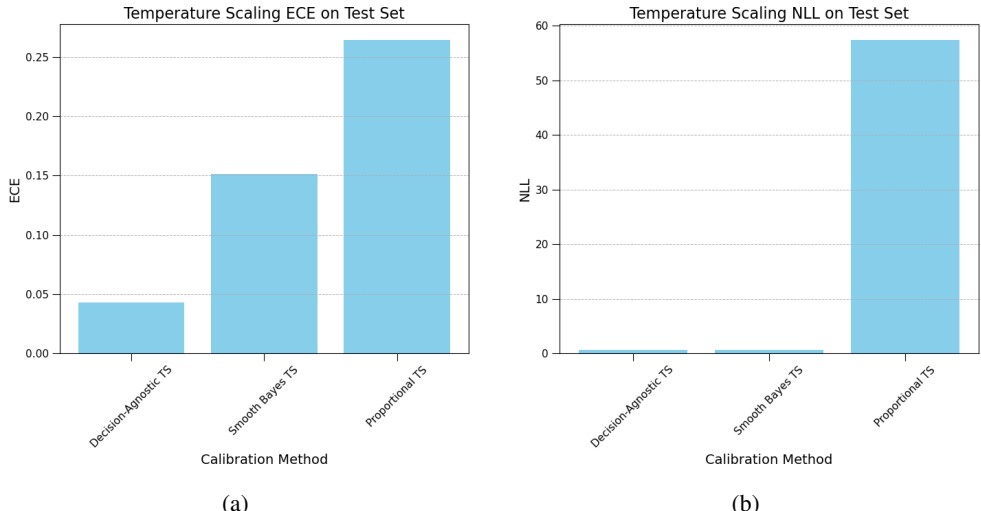

(a)                                      (b)

Figure 6: Temperature scaling calibration performance at test time: (a) expected calibration error (ECE); (b) negative-log likelihood (NLL)

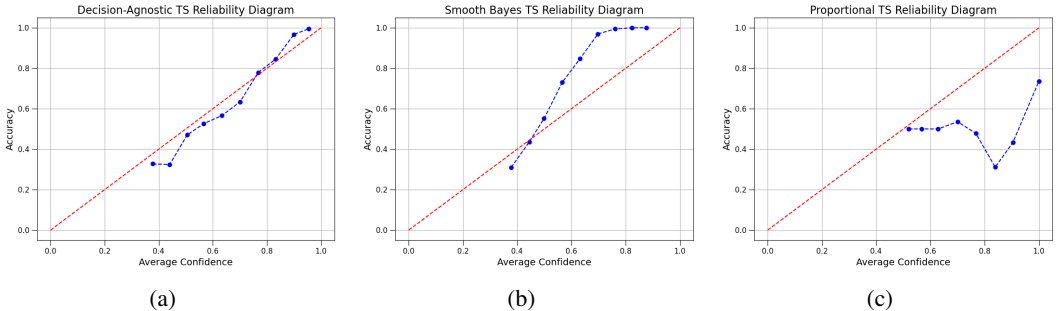

(a)                                      (b)                                      (c)

Figure 7: Reliability diagrams at test time for each calibration method: (a) decision-agnostic; (b) Smooth Bayes; (c) Proportional.

