# OpenReview forum: "Decision-Driven Calibration for Cost-Sensitive Uncertainty Quantification"
_NeurIPS.cc/2024/Workshop/BDU — NeurIPS BDU Workshop 2024 Poster_

### Official Review · Reviewer_ZP8c · 2024-09-21
**Comment for Decision-Driven Calibration for Cost-Sensitive Uncertainty Quantification**

**Rating:** 5
**Confidence:** 4

**Review:**

**Pros**:
1. The paper introduces a novel and interesting approach by considering the direct influence of Uncertainty Quantification (UQ) on the subsequent actions taken by a downstream decision-maker. This perspective adds valuable insight to the field and worths further exploration.

**Cons**:
1. There are several technical errors in the paper that need revision. For example:
   - On page 2, line 58: The notation $X = ℝ^d$ should be corrected to $X ∈ ℝ^d$.
   - On page 4, line 139: The model name "ViT_B_16" should be revised to "ViT-B/16" in accordance with Dosovitskiy et al. [1].

2. On page 3, line 96, the paper uses $σ(z)$ for approximating the $argmax$, but it is strongly recommended to introduce a temperature parameter. This would help control the approximation effect, as a lower temperature results in a closer approximation (and lower gradient flow).

3. The paper lacks a conclusion section, which is crucial for summarizing the key findings and contributions. Additionally, Figure 1 requires significant improvement. For instance, the text on the X-axis could be formatted into multiple rows for better readability.

4. It is strongly suggested that the paper includes a discussion on the differences between this decision-driven approach and policy gradient methods. The expected decision cost formulation on page 2, line 74, shares similarities with policy gradient, and addressing this connection would provide greater clarity and depth to the proposed method.

---

**Reference**:
[1] A. Dosovitskiy, L. Beyer, A. Kolesnikov, D. Weissenborn, X. Zhai, T. Unterthiner, M. Dehghani, M. Minderer, G. Heigold, S. Gelly, J. Uszkoreit, and N. Houlsby, “An image is worth 16x16 words: Transformers for image recognition at scale,” in *International Conference on Learning Representations*, 2021.

---

### Official Review · Reviewer_oaRo · 2024-09-28
**Great idea, but format and presentation can be improved**

**Rating:** 6
**Confidence:** 3

**Review:**

Summary:
The paper proposes a new decision-driven calibration to directly optimize UQ to minimize incurred costs.

Strengths:
1. Strong technical explanation of the underlying concept.
2. The objective is clearly stated and justified.
3. The examples are appropriate and of high relevance for real-world applications.

Weaknesses:
1. Many sentences are too long and wordy. It is difficult to follow the logical flow of the idea. My recommendation would be to break any long sentence into smaller ones that comprise only one idea.
2. The format of the paper is sometimes lacking clarity. For example, there is no clear delineation to where the conclusion starts and where the results end.
3. The paper is limited in visualisations. A higher number and better visuals might aid the reader in following the paper better.

Overall, it is a good idea, but a hard to read paper.

---

### Decision · Program_Chairs · 2024-10-09

Accept (Poster)